# Biofilm Formation, c-di-GMP Production, and Antimicrobial Resistance in Staphylococcal Strains Isolated from Prosthetic Joint Infections: A Pilot Study in Total Hip and Knee Arthroplasty Patients

**DOI:** 10.3390/ijms26188929

**Published:** 2025-09-13

**Authors:** Andrea Liberatore, Alessia Bertoldi, Alice Balboni, Liliana Gabrielli, Alessia Cantiani, Federica Lanna, Maria Sartori, Silvia Brogini, Gianluca Giavaresi, Tiziana Lazzarotto

**Affiliations:** 1Microbiology Unit, IRCCS Azienda Ospedaliero-Universitaria di Bologna, 40138 Bologna, Italy; andrea.liberatore@studio.unibo.it (A.L.); alessia.bertoldi3@unibo.it (A.B.); tiziana.lazzarotto@unibo.it (T.L.); 2Microbiology, Department of Medical and Surgical Sciences, University of Bologna, 40138 Bologna, Italy; alice.balboni2@studio.unibo.it (A.B.); alessia.cantiani@studio.unibo.it (A.C.); federica.lanna@aosp.bo.it (F.L.); 3Surgical Sciences and Technologies, IRCCS Istituto Ortopedico Rizzoli, 40136 Bologna, Italy; maria.sartori@ior.it (M.S.); silvia.brogini@ior.it (S.B.); gianluca.giavaresi@ior.it (G.G.)

**Keywords:** prosthetic joint infection, *Staphylococcus aureus*, biofilm formation, c-di-GMP, antimicrobial resistance

## Abstract

Total joint arthroplasty (TJA) and total joint replacement (TJR) are effective treatments for end-stage osteoarthritis, but prosthetic joint infections (PJIs) remain a significant complication. These infections are often associated with bacteria that form biofilms, which contribute to their persistence and resistance to treatment. The aim of this study was to investigate the biofilm-forming ability, cyclic diguanylic acid (c-di-GMP) production, and the presence of biofilm-associated genes in *Staphylococcus aureus* and coagulase-negative *Staphylococci* (CoNS) isolates obtained from synovial fluid samples of patients with PJIs following TJA and TJR. A total of 198 samples were analyzed, with bacterial growth detected in 33 samples (16.7%). Among these, 10 strains of *S. aureus* and 22 strains of CoNS were identified. Biofilm formation was evaluated using the crystal violet assay, and c-di-GMP levels were measured. A statistically significant linear regression was found between biofilm formation and c-di-GMP production (*p* = 0.016, R^2^ = 0.18). Genetic analysis revealed the presence of biofilm-associated genes, including *icaA*, *clfA*, *fnbA* in *S. aureus*, and *atlE*, *fbe* in CoNS. Furthermore, there was a statistically significant difference in c-di-GMP production between strains harboring the icaA gene and strains without icaA (*p* = 0.016), while oxacillin resistance was detected more frequently in strains carrying fbe gene (*p* = 0.031). The study emphasizes the variability in antibiotic resistance profiles among staphylococcal isolates, underscoring the complexity of managing these infections.

## 1. Introduction

Total joint arthroplasty (TJA) and total joint replacement (TJR) are well-established and highly effective surgical treatments for end-stage osteoarthritis, significantly improving patients’ quality of life [1,2]. However, prosthetic joint infection (PJI)—an infection involving the prosthetic joint and surrounding soft tissues—remains one of the most serious complications in orthopedic surgery. Accurate microbiological diagnosis is essential for effective clinical management of such infections. Although the incidence of PJI is relatively low, ranging from 0.1% to 4.0% according to literature [3,4], its clinical impact is significant. Studies have reported a 1-year mortality rate of up to 8% following the index procedure and a twofold increase in in-hospital mortality with each surgical admission [5,6,7].

In response to the COVID-19 pandemic, several countries reported a reduction in elective orthopedic surgeries in 2020 [8]. In Italy, for example, hip replacements decreased by 20% and knee replacements by approximately 30% compared to 2019 [8]. Despite this global decline in procedures, the incidence of periprosthetic infections has shown a rising trend and is now recognized as one of the most challenging complications in modern orthopedics. Several patient-related factors have been associated with an increased risk of PJI, including comorbidities such as rheumatoid arthritis, diabetes mellitus, malignancy, chronic kidney disease, and obesity [9,10]. Bloodstream infection (BSI) in the presence of a prosthetic joint is a well-established risk factor for PJI, particularly in cases involving *Staphylococcus* spp., with studies indicating that 25% to 34% of staphylococcal BSIs can lead to PJI [11,12].

The most common microorganisms responsible for PJIs are *Staphylococcus* spp., particularly *Staphylococcus aureus* and coagulase-negative *Staphylococci* (e.g., *Staphylococcus epidermidis*) [13]. However, the microbial etiology can vary significantly based on factors such as time since implantation, joint site, and geographic location; for instance, *S. aureus* predominates in the U.S., whereas *S. epidermidis* is more common in Europe [13]. Identification of the causative pathogen through culture is crucial for guiding antimicrobial therapy. Culture positivity ranges from 65% to 94% of PJIs, with acute infections often yielding positive cultures in up to 90% of cases [14]. Early-onset PJIs require precise microbiological diagnosis due to their frequent management via surgical debridement [15,16,17].

Treatment is further complicated when virulent and antimicrobial-resistant pathogens are involved, such as methicillin-resistant *S. aureus* (MRSA), rifampin-resistant *Staphylococci*, or ciprofloxacin-resistant Gram-negative bacilli, all of which are associated with poorer outcomes [18,19]. Culture-negative PJIs commonly result from prior antibiotic exposure or the presence of fastidious organisms such as fungi or mycobacteria. Advances in molecular diagnostics—including broad-range polymerase chain reaction (PCR) and metagenomic shotgun sequencing—are improving the detection of these difficult-to-identify pathogens [20].

A major factor contributing to the persistence of PJIs is the formation of bacterial biofilms. Surgical implants and prosthetic materials serve as abiotic surfaces conducive to microbial adhesion and biofilm development. In biofilms, bacteria aggregate into a sessile community embedded within a self-produced extracellular polymeric substance (EPS) [21,22]. Biofilms confer significant resistance to antibiotics and host immune defenses [23], in part due to altered gene expression, reduced antimicrobial penetration, and the protective properties of the EPS matrix [24,25,26]. Classical culture techniques often fail to detect biofilm-associated pathogens, with success rates as low as 30% [27]. Therefore, effective biofilm disruption is necessary prior to standard microbiological analysis [28]. Gram-positive cocci—particularly *S. aureus*, coagulase-negative *Staphylococci* (CoNS), and enterococci—account for approximately 75% of biofilm-associated PJIs [13,29,30].

Despite ongoing efforts, standardized guidelines for the diagnosis and management of biofilm-related infections remain incomplete and controversial. Given that many medical devices (e.g., catheters, implants) are prone to biofilm colonization [31], there is an urgent need for reliable assays to detect biofilm-forming strains. In this context, the secondary messenger bis(3′,5′)-cyclic di-guanylic acid (c-di-GMP) plays a key regulatory role in bacterial physiology. It governs various cellular processes, including virulence and biofilm formation [32]. Elevated intracellular levels of c-di-GMP promote a transition from a motile, planktonic lifestyle to a sessile, biofilm-forming state [33], partly by inducing adhesin and exopolysaccharide biosynthesis while inhibiting flagellar motility [34]. Furthermore, c-di-GMP can regulate gene expression post-transcriptionally by interacting with mRNA or small regulatory RNAs [35].

Several genes have been implicated in staphylococcal biofilm formation. Among them, the *ica* operon is well-characterized for encoding enzymes responsible for synthesizing polysaccharide intercellular adhesin (PIA) [36]. More recently, proteinaceous components have also been shown to contribute significantly to biofilm development [37]. Surface adhesins such as fibronectin-binding proteins (FnBPs) and clumping factor A (ClfA) are involved in host tissue colonization and immune evasion. Notably, ClfA binds complement factor I, facilitating neutrophil evasion [38], while FnBPs and elastin bind plasminogen, enhancing biofilm formation [39,40]. In CoNS, biofilm formation involves surface proteins such as autolysin E (AtlE) and the fibrinogen-binding protein of *S. epidermidis* (Fbe), in addition to PIA production via the *ica* operon [41].

The aim of this study was to characterize the microbial population identified in patients with PJI (≥90 days post-implantation) after TJA and TJR, focusing on c-di-GMP production and the activation of specific genes, like *ica*, as previously described, in order to assess the actual ability of microbial populations—particularly *S. aureus* and CoNS—to form biofilms. Biofilm formation represents a negative prognostic factor that significantly complicates both treatment and infection eradication. Therefore, accurate and timely identification of biofilms may represent a crucial additional tool for the appropriate management of therapy, not only guiding the selection of the most effective antibiotics but also modulating treatment duration and therapeutic strategies aimed at improving infection control and patient outcomes.

## 2. Results

### 2.1. Bacteria Identification and Antibiotic Susceptibility

In this study, a total of 198 synovial fluid samples were collected and sent to the Microbiology Unit at the IRCCS Azienda Ospedaliero-Universitaria of Bologna for microbiological analysis. Among these, 48 (24.2%) samples were obtained from patients with clinically suspected PJ infectionetiology, and 150 (75.8%) were from prosthetics with clinically suspected non-infected etiology. Bacterial growth was detected in 24/48 samples (50%) from prosthetics with clinically suspected infection, while among the 150 clinically suspected non-infected samples, bacterial growth was observed in 9 samples (6%). Overall, in 33/198 samples (16.7%) we found bacterial growth, including 10 strains of *S. aureus*, 22 strains of CoNS, 1 strain of *Enterococcus faecalis*, 1 strain of *Corynebacterium striatum*, and 1 strain of *Micrococcus luteus* (Figure 1). Co-infections were observed in two cases: one involving *S. epidermidis* and *E. faecalis*, associated with a prosthesis with infectious etiology, and one involving a strain of *S. epidermidis* and a strain of *Staphylococcus capitis*, associated with a prosthesis with non-infectious etiology. Our study focused on *Staphylococcus* strains; other strains were excluded from further analysis.

Table 1 shows the susceptibility profiles of all *Staphylococcus* strains.

Among the 10 *S. aureus* isolates, 30% (3/10) demonstrated resistance to three classes of antimicrobial agents, classifying them as multidrug-resistant (MDR) isolates. All MDR strains were MRSA. The resistance rates among the 10 *S. aureus* isolates showed significant variability. In particular, resistance to erythromycin was observed in 60% of the isolates, while resistance to oxacillin, levofloxacin, and clindamycin was below 30%. All *S. aureus* isolates tested were susceptible to tetracycline, vancomycin, trimethoprim-sulfamethoxazole, teicoplanin, and daptomycin.

Among the 22CoNS isolates, 27.3% (6/22) were characterized as MDR. This group included 5 (83.3%) oxacillin-resistant strains and 1 (16.7%) oxacillin-susceptible strains. Resistance rates among the 22 CoNS isolates showed substantial variation. Resistance to oxacillin, erythromycin, and levofloxacin was observed in over 30% of the isolates, while resistance to clindamycin, tetracycline, and trimethoprim-sulfamethoxazole was below 19%. All CoNS isolates were susceptible to vancomycin, teicoplanin, and daptomycin.

### 2.2. Biofilm-Forming Ability

The biofilm formation ability of the different bacterial strains was further assessed using the crystal violet assay. Biofilm formation was observed in all isolates (32/32) (Figure 2). The assay demonstrated high reproducibility, with only minor differences observed between the replicates.

Among the 10 *S. aureus* isolates, poor biofilm formation was observed in 10% (1/10), weak biofilm formation in 40% (4/10), moderate biofilm formation in 20% (2/10), and strong biofilm formation in 30% (3/10) of isolates (Table 2).

Among the 22 CoNS isolates, poor biofilm formation was observed in 9.1% (2/22), weak biofilm formation in 13.6% (3/22), moderate biofilm formation in 40.9% (9/22), and strong biofilm formation in 36.4% (8/22) (Table 3). Within the group displaying moderate and strong biofilm formation (Figure 2), the majority of isolates (55.6%, 5/9 and 62.5%, 5/8, respectively) were *S. epidermidis*.

No statistically significant difference was found between the ability to form biofilms between bacterial strains from prosthesis with infectious and non-infectious etiology (*p* > 0.05).

### 2.3. Concentration of c-di-GMP

The concentration of c-di-GMP was measured in all bacterial isolates (32/32) (Table 4). The levels of c-di-GMP were then compared with the results from the crystal violet assay. The linear regression between biofilm and c-di-GMP was statistically significant (*p* = 0.016, R^2^ = 0.18) (Figure 3).

### 2.4. Genes Involved in Biofilm Formation

All *Staphylococcus* strains were tested for various genes involved in biofilm formation. Among the 10 *S. aureus* isolates, 100% (10/10) were positive for the presence of the *icaA* and *clfA* genes, while 60% (6/10) exhibited the *fnbA* gene (Table 5). Among the 22 CoNS isolates, the *icaA* gene was observed in 31.8% (7/22), the *atlE* gene in 63.6% (14/22), and the *fbe* gene in 59.1% (13/22) (Table 6).

Subsequently, a significant difference was observed between the presence of the *icaA* gene and c-di-GMP concentration (434.7 ± 151.3 vs. 572.1 ± 179.6; *p* = 0.016) (Figure 4). Interestingly, strains with the *fbe* gene were more frequently associated with oxacillin resistance compared to those without the *fbe* gene (6.5 ± 2.1 vs. 4.5 ± 4.9; *p* = 0.031) (Figure 5).

## 3. Discussion

Infection as a consequence of joint replacement is a rare event, but it can lead to severe complications due to the presence of pathogens capable of forming biofilms. However, with the increasing life expectancy, surgical procedures requiring total joint replacement are becoming more common, resulting in an alarming rise in difficult-to-treat (peri) PJIs [42]. It is widely acknowledged that the presence of microbial biofilms is a major virulence factor associated with challenging infections. Bacteria embedded in biofilms are inherently more resistant to environmental and chemical agents, tolerating higher concentrations of antibiotics than their planktonic counterparts, allowing them to survive long-term in natural ecosystems and animal hosts [43]. Moreover, biofilm formation can prevent bacterial clearance and hinder the host immune response, enabling bacteria to invade host tissues and increase morbidity and mortality [36,44]. Therefore, the aim of the present study was to better characterize biofilm-associated determinants of virulence in different bacterial strains isolated from synovial fluid derived from PJIs, with particular focus on factors influencing biofilm formation rather than antimicrobial resistance mechanisms.

In this ongoing study, the prevalence of PJI from joint replacements was found to be 16.7% (33/198). Notably, in all synovial fluids that exhibited bacterial growth, only gram-positive bacteria were isolated (100%, 33/33). Incidence rates for PJIs can vary between 0.1% and 20%, depending on the specific orthopedic subspecialties. However, for hip and knee arthroplasty, the incidence of surgical site infections typically ranges from 0.1% to 4.0% [45,46,47]. It is important to note that, when specifically considering *S. aureus*, only 10 synovial fluids tested positive for this pathogen, yielding an incidence of 5.2% (10/193). These findings are consistent with a 17-year study conducted in Finland, which observed a similar range of *S. aureus* infections following hip and knee arthroplasty, with an incidence varying from 1.6% to 4.0% [46].

Antimicrobial susceptibility of the staphylococcal isolates was assessed by determining the MIC, and the methicillin resistance status was evaluated through oxacillin MIC testing. Overall, methicillin resistance was found in 37.5% (12/32) of bacterial isolates, while 100% (32/32) of the strains were vancomycin-susceptible. Although the number of *S. aureus* isolates was limited, the MRSA incidence rate was 30% (3/10). This rate is consistent with a study by Pimentel de Araujo et al. (2022), which reported an incidence of MRSA in osteomyelitis cases of 30.1% [48]. No correlation was found between the identification of methicillin-resistant isolates and their ability to produce biofilm or c-di-GMP. This lack of correlation is most likely due to the relatively small number of isolates analyzed, which limits the statistical power of our observations. Nevertheless, literature indicates that the presence of biofilm itself may offer effective protection from antibiotic action, as bacteria embedded in biofilms can survive antibiotic concentrations 10 to 10,000 times greater than their planktonic counterparts [43,49].

Subsequently, we aimed to further phenotypically characterize the isolated strains based on their ability to produce biofilm. The crystal violet assay, a widely used method for biofilm quantification, classifies bacterial strains as poor, weak, moderate, or strong biofilm producers [50,51]. Additionally, the production of c-di-GMP, a second messenger known to regulate bacterial physiology, including biofilm formation, was assessed [52,53]. Most of the isolated strains were classified as strong (34.4%, 11/32) or moderate (34.4%, 11/32) biofilm producers. Although a statistically significant linear regression (*p* < 0.05) was observed between biofilm production and c-di-GMP levels, consistent with findings in other bacteria such as *Pseudomonas aeruginosa*, *Pseudomonas resinovorans*, and *Vibrio vulnificus* [54,55,56], the low R^2^ value (0.20) indicates that c-di-GMP explains only a small proportion of the variability in biofilm formation. This suggests that, while the association is real, c-di-GMP alone is a poor predictive biomarker for biofilm production and likely requires consideration in combination with other regulatory factors.

Biofilm formation is a complex process that involves numerous genes, proteins, and signaling pathways. To better understand the genetic background of the isolated strains, we investigated the presence of several biofilm-associated genes. Specifically, all bacterial isolates were tested for the presence of the *icaA* gene, which is responsible for the synthesis of the extracellular matrix and biofilm formation. The presence of the *icaA* gene was detected in 53.1% (17/32) of bacterial isolates. Among CoNS, 31.8% (7/22) of isolates carried the *icaA* gene, while in all *S. aureus* isolates (100%, 10/10), the *icaA* gene was present. A study by Pimentel de Araujo et al. (2022) reported that 95% of *S. aureus* isolates responsible for osteomyelitis carried the *icaA* gene [48]. Moreover, a significant difference in intracellular c-di-GMP concentration was observed between *icaA*-positive and *icaA*-negative strains. While this finding suggests a link between gene presence and signaling activity, it also highlights the complexity of the regulatory mechanisms involved.

Furthermore, we observed a statistically significant difference in c-di-GMP intracellular concentration between strains harboring the *icaA* gene and strains without the *icaA* gene.

We also examined *S. aureus* isolates for the presence of *fnbA* and *clfA*. Both genes are known to contribute to tissue colonization and biofilm formation [57,58]. All *S. aureus* isolates tested positive for the *clfA* gene (100%, 10/10), whereas 60% (6/10) carried the *fnbA* gene. These findings are consistent with studies showing a high frequency of *clfA* in *S. aureus* isolates from hematogenous infections [59]. However, Kouidhi et al. (2010) reported a much lower frequency of *clfA* (9.1%) in *S. aureus* strains isolated from dental caries, suggesting that the distribution of *clfA* may vary depending on the infection site [60].

For CoNS, we investigated the presence of the *atlE* and *fbe* genes. The *atlE* gene codes for an autolysin that mediates cell wall lysis, leading to DNA release into the biofilm matrix and contributing to irreversible adhesion during biofilm development [61,62]. The *fbe* gene encodes a fibrinogen-binding protein involved in attachment and biofilm production [63,64]. The *atlE* gene was detected in 63.6% (14/22) of CoNS isolates, and the *fbe* gene was found in 59.1% (13/22). Interestingly, strains with the *fbe* gene were more frequently associated with oxacillin resistance compared to those without the *fbe* gene (*p* < 0.05). The involvement of fbe in biofilm formation could explain the statistically significant difference between the groups.

While few studies have addressed this specific aspect, our findings are in line with previous research suggesting that biofilm-associated genes are more frequently carried by methicillin-resistant CoNS strains compared to methicillin-sensitive strains [65,66].

Although biofilm formation was significantly associated with c-di-GMP production in this cohort, the low R^2^ value (0.18) raises doubts about the consistency of the relationship, suggesting that further studies are needed to better characterize this pathway. Infact, as reported by Zhu et al., no correlation between biofilm formation and intracellular c-di-GMP levels was observed in *S. epidermidis* [67].

## 4. Materials and Methods

### 4.1. Study Population

A prospective ongoing clinical study was conducted at the IRCCS Istituto Ortopedico Rizzoli in Bologna (Italy), in collaboration with the Microbiology Unit of the IRCCS Azienda Ospedaliero-Universitaria of Bologna. The study aimed to enroll patients scheduled for hip or knee prosthesis revision due to late infection (beyond 90 days)and to prosthetic failure not related to infection. The protocol was approved by the Ethics Committee of Area Vasta Emilia Centro (CE-AVEC; protocol number 37/2021/Sper/IOR) and the study was conducted in accordance with the Declaration of Helsinki (ClinicalTrial.gov: 2021-04-26 ID: NCT04858217). Written informed consent was obtained from all participants. Between February 2022 and January 2025, 203 patients were enrolled. Inclusion criteria were (1) male and female patients aged ≥ 18 years; (2) patients scheduled for hip or knee prosthesis revision surgery after at least 90 days from the date of the primary arthroplasty due to (a) late PJI, or (b) non-infective causes (loosening, wear, instability, malalignment, adverse local tissue reactions, or other aseptic conditions) in patients who had not undergone previous re-operations on the same joint and whose revision procedure was planned as a single-stage intervention; (3) availability of previous clinical data as well as laboratory and radiological examinations. Exclusion criteria were (1) patients with early PJI with a clinical latency of symptoms of less than 90 days; (2) patients with PJI involving joints other than the hip or knee; (3) patients with severe cognitive impairment or psychiatric disorders; (4) pregnant women. During recruitment, patients were classified as “infected” according to the Musculoskeletal Infection Society (MSIS) major criteria [68], defined by either (1) a sinus tract communicating with the joint or (2) a positive microbiological culture from at least two separate periprosthetic tissue/fluid samples or, alternatively, if they met at least three of minor criteria. 

### 4.2. Bacterial Isolates and Growth Conditions

Prior to antibiotic administration, 5 mL of synovial fluid were aseptically collected and anonymized. A 1.5 mL aliquot was immediately transferred to the microbiology unit for analysis. From each sample, 10 µL were cultured on tryptic soy agar supplemented with 5% sheep blood (TSA with sheep blood, Thermo Fisher Scientific, Milano, Italy), and 20 µL were inoculated into brain heart infusion broth (BHI, Liofilchem S.r.l., Teramo, Italy) to enhance bacterial growth. The remaining sample was stored at −80 °C. Agar plates were incubated at 37 °C and examined for aerobic bacteria growth at 24 and 48 h. BHI cultures were incubated at 37 °C and monitored daily for up to 15 days. In the absence of visible turbidity, 10 µL of broth were subcultured on solid media and incubated at 37 °C for an additional 48 h. Colonies were identified by MALDI-TOF MS (Bruker Daltonics, Leipzig, Germany). Antimicrobial susceptibility testing was performed using the MicroScan WalkAway-96 system (Beckman Coulter, Brea, CA, USA). Minimum inhibitory concentrations (MICs) were interpreted according to EUCAST breakpoints v15.0 (http://www.eucast.org/clinical_breakpoints/, accessed on 4 February 2025).

### 4.3. Crystal Violet Assay

Biofilm formation was assessed using a crystal violet staining protocol adapted from Di Domenico et al. (2016) [69]. Briefly, 200 µL of bacterial suspension (10^7^ CFU/mL in BHI) were added to each well of a 96-well polystyrene plate and incubated at 37 °C for 24 h without agitation. Each strain was tested in triplicate. Two hundred µL of BHI, dispensed in triplicate, were used as a negative control. Wells were gently washed three times with distilled water and air-dried for 45 min. Adherent biofilms were stained with 0.1% crystal violet solution (Delcon S.r.l., Bergamo, Italy) for 20 min, rinsed four times, and the retained stain was solubilized using 200 µL of ethanol/acetone (80:20). Absorbance was measured at 550 nm (OD_550_) using a microplate reader (ThermoLabsystemsMultiskan Ascent, Thermo Fisher Scientific, Waltham, MA, USA). Biofilm production was categorized, based on the absorbance measured, as poor, weak, moderate, or strong, following the semi-quantitative criteria established by Di Domenico et al. (2016) [69]. The cut-off optical density (ODc) was defined as three standard deviations above the mean OD of the negative control. Strains were classified as follows:Poor: OD < ODcWeak: ODc < OD < 2 × ODcModerate: 2 × ODc < OD < 4 × ODcStrong: OD > 4 × ODc

### 4.4. Quantification of Cyclic di-GMP

The intracellular concentration of c-di-GMP, a bacterial second messenger associated with biofilm regulation, was quantified using the Cyclic di-GMP ELISA Kit (Cayman Chemical, Ann Arbor, MI, USA) starting from a bacterial suspension of 3.6 × 10^9^ CFU/mL and following the manufacturer’s instructions. The bacterial strain extraction procedure was performed following the instructions “B-PER^®^ Bacterial Protein Extraction Reagent” (Thermo Fisher Scientific, Milano, Italy) as indicated in the Cyclic di-GMP ELISA kit document. Absorbance was measured at 450 nm (OD_450_). Data analysis and normalization were performed using the Cayman spreadsheet software provided in the kit.

### 4.5. PCR Analysis of Biofilm-Associated Genes

Chromosomal DNA was extracted using the ELITe InGenius system (ELITech Group, Torino, Italy) according to the manufacturer’s instructions. PCR was performed using the GoTaq^®^ G2 Hot Start Polymerase Kit (Promega, Madison, WI, USA). The presence of the *icaA* gene (GenBank accession no. AF086783) was assessed using icaA-F and icaA-R primers (Table 7) [50]. Each 50 µL PCR reaction included 10 µL 5× Green GoTaq Flexi Buffer, 2.5 mM MgCl_2_, 1 µM of each primer, 1 µL dNTPs (10 mM), 1.25 U polymerase, 5 µL DNA, and nuclease-free water. Thermal cycling conditions were as follows: initial denaturation at 95 °C for 2 min; 30 cycles of 95 °C for 30 s, 60 °C for 30 s, 72 °C for 15 s; final extension at 72 °C for 5 min. PCR products were resolved on 1% agarose gel; a 670 bp band indicated *icaA* positivity.

*S. aureus* strains were further tested for *fnbA* (GenBank: J04151) and *clfA* (GenBank: Z18852) using specific primers (Table 7) [50]. For *fnbA*, the master mix contained 1.5 mM MgCl_2_; for *clfA*, 3 mM MgCl_2_ was used. PCR conditions were as follows: 95 °C for 2 min; 30 cycles of 95 °C for 30 s, 53 °C for 30 s, 72 °C for 90 s; final extension at 72 °C for 5 min. Expected amplicons were 1360 bp (*fnbA*) and 1580 bp (*clfA*).

CoNS strains were analyzed for *atlE* (GenBank: U71377.1) and *fbe* (GenBank: Y17116.1) using atlE-F/atlE-R and fbe-F/fbe-R primers (Table 7) [50]. PCR reactions included 1.5 mM MgCl_2_ and 0.2 µM of each primer in a 50 µL volume. Thermocycling was as follows: 95 °C for 2 min; 30 cycles of 95 °C for 30 s, 62 °C for 30 s, 72 °C for 30 s; final extension at 72 °C for 5 min. Amplicon sizes were 680 bp for *atlE* and 270 bp for *fbe*.

### 4.6. Statistical Analysis

Categorical variables were reported as absolute values and percentage frequencies. Data distribution was assessed by Kolmogorov–Smirnov Test. Linear regression, unpaired two-tailed Mann–Whitney test, and contingency tables were used to evaluate statistically significant differences. A *p*-value of less than 0.05 was considered statistically significant. Statistical analyses were performed using GraphPad Prism version 8.0.1 for Windows.

## 5. Conclusions

There are some limitations to the current study. First, the number of isolates was relatively small. Moreover, biofilm formation represents a complex phenomenon governed by the involvement of numerous genes, proteins, and signaling pathways, many of which remain only partially characterized. Due to the complexity of this process, the present study has restricted its investigation to a subset of key factors known to be associated with biofilm formation.

Future research directions include a substantial increase in the number of samples analyzed and a more comprehensive examination of the expression patterns of biofilm-associated genes. Such approaches are expected to provide deeper insights into the genetic background of the bacterial strains and enable a more precise classification of isolates based on their biofilm-producing capacity.

## Figures and Tables

**Figure 1 ijms-26-08929-f001:**
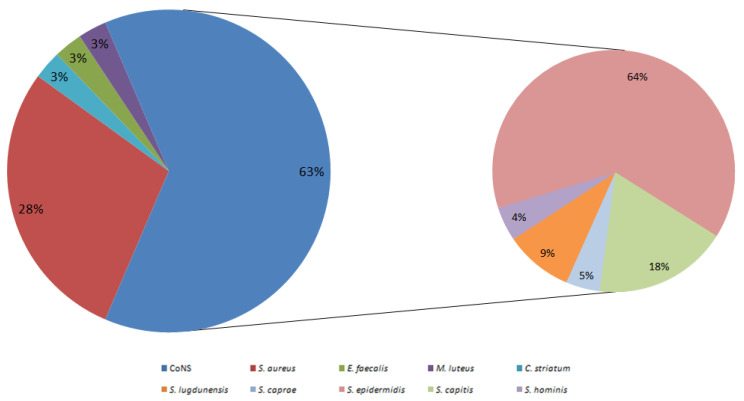
Distribution of bacterial isolates. Left: overall composition—coagulase-negative *staphylococci* (CoNS), *Staphylococcus aureus*, *Enterococcus faecalis*, *C. striatum* and *M. luteus* (percentages as shown). Right (inset): breakdown of CoNS isolates only, comprising *S. hominis*, *S. lugdunensis*, *S. caprae*, *S. epidermidis*, and *S. capitis*. Distribution of bacterial isolates. **Left**: overall composition—coagulase-negative *staphylococci* (CoNS), *Staphylococcus aureus*, *Enterococcus faecalis*, *C. striatum* and *M. luteus* (percentages as shown). **Right** (inset): breakdown of CoNS isolates only, comprising *S. hominis*, *S. lugdunensis*, *S. caprae*, *S. epidermidis*, and *S. capitis*.

**Figure 2 ijms-26-08929-f002:**
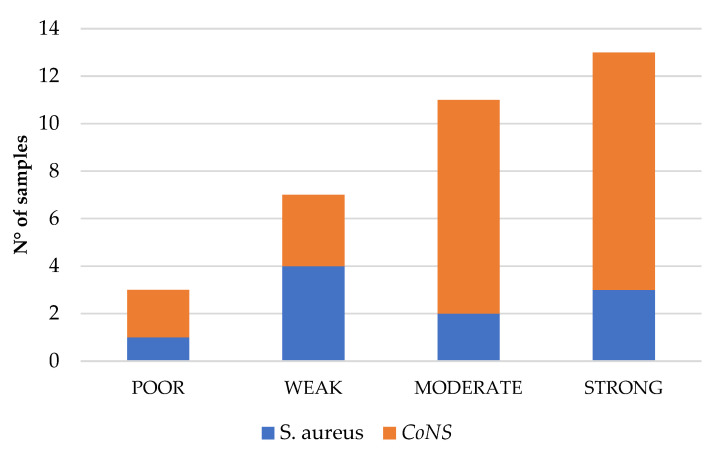
Biofilm formation ability of *Staphylococcus* strains evaluated by crystal violet assay. Strains show variability from poor to strong, with a prevalence of *S. epidermidis* among moderate/strong producers.

**Figure 3 ijms-26-08929-f003:**
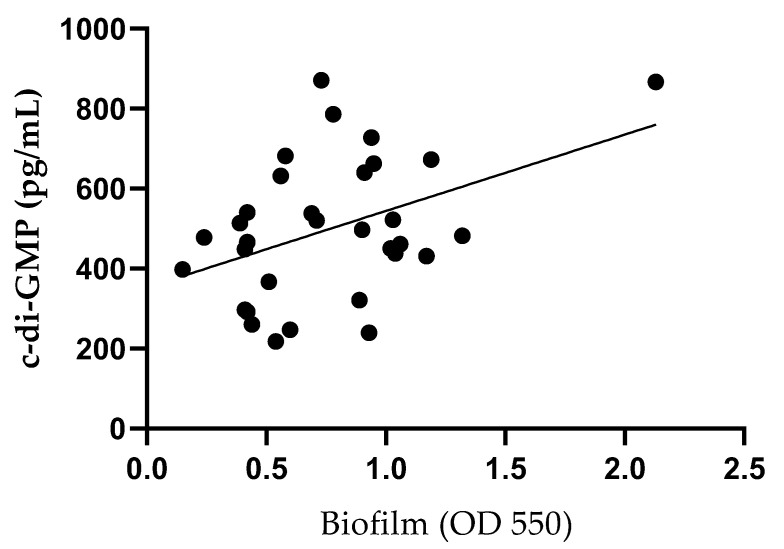
Linear regression between biofilm-forming capacity and c-di-GMP production. Despite a significant difference (*p* = 0.016), the low R^2^ (R^2^ = 0.18) indicates that other factors contribute to the biofilm production.

**Figure 4 ijms-26-08929-f004:**
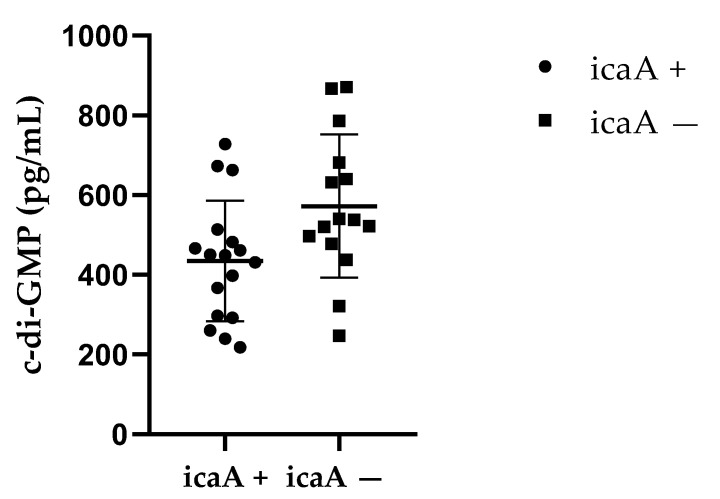
Comparison of c-di-GMP production in *icaA*+ and *icaA*− strains. Negative strains showed significantly higher concentrations (*p* = 0.016), indicating multifactorial regulation of biofilm formation.

**Figure 5 ijms-26-08929-f005:**
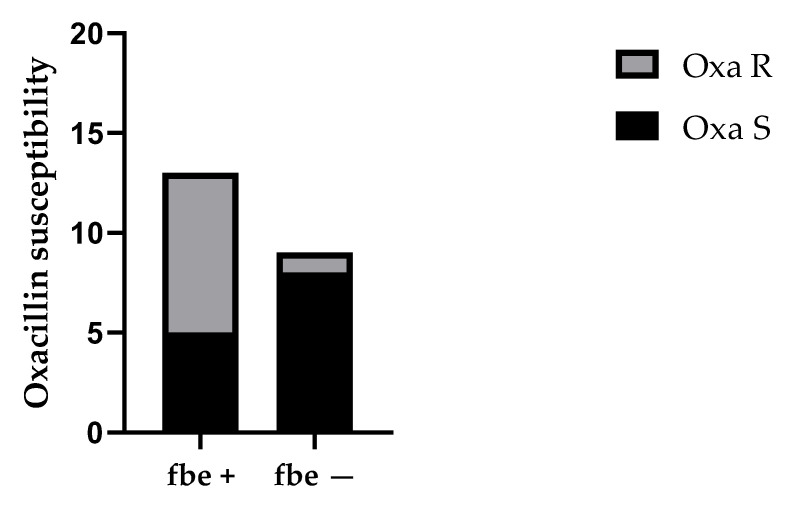
Analysis of oxacillin susceptibility in *fbe*+ and *fbe*− strains. Positive strains showed a significantly higher frequency of oxacillin resistance (*p* = 0.031).

**Table 1 ijms-26-08929-t001:** Susceptibility profiles of Staphylococcal isolates. Abbreviations: S—susceptible; I—susceptible, increased exposure; R—resistant; OXA—oxacillin; ERY—erythromycin; LEV—levofloxacin; CD—clindamycin; TET—tetracycline; SXT—trimethoprim/sulfamethoxazole; Van—vancomycin; TEI—teicoplanin; DAP—daptomycin; N.A.—not analyzed.

Isolates	Prosthetics Etiology	Antibioticdrugs								
		OXA	ERY	LEV	CD	TET	SXT	VAN	TEI	DAP
*S. epidermidis*	Infected	R	S	I	S	S	S	S	S	S
*S. aureus*	Infected	S	R	I	S	S	S	S	S	S
*S. capitis*	Infected	S	S	I	S	S	S	S	S	S
*S. aureus*	Infected	S	R	I	S	S	S	S	S	S
*S. epidermidis*	Infected	R	S	I	S	S	S	S	S	S
*S. capitis*	Infected	S	S	I	S	S	S	S	S	S
*S. aureus*	Non-infected	R	R	R	S	S	S	S	S	S
*S. aureus*	Infected	R	R	R	S	S	S	S	S	S
*S. lugdunensis*	Infected	S	S	I	S	S	S	S	S	S
*S. aureus*	Infected	S	S	I	S	S	S	S	S	N.A.
*S. epidermidis*	Infected	R	S	R	S	S	I	S	S	S
*S. aureus*	Infected	S	R	I	R	S	S	S	S	N.A.
*S. epidermidis*	Infected	R	R	N.A.	R	S	R	S	S	S
*S. epidermidis*	Infected	R	R	R	R	S	R	S	S	S
*S. aureus*	Infected	S	S	I	S	S	S	S	S	S
*S. epidermidis*	Non-infected	S	S	R	S	R	S	S	S	S
*S. epidermidis*	Infected	R	R	R	R	S	S	S	S	S
*S. epidermidis*	Infected	R	S	R	S	S	R	S	S	S
*S. epidermidis*	Infected	R	R	I	S	S	S	S	S	S
*S. capitis*	Infected	R	R	R	S	S	S	S	S	S
*S. epidermidis*	Non-infected	R	R	I	S	S	S	S	S	S
*S. epidermidis*	Non-infected	S	S	R	S	S	R	S	S	S
*S. epidermidis*	Infected	S	R	R	S	R	S	S	S	S
*S. aureus*	Infected	S	S	I	S	S	S	S	S	S
*S. lugdunensis*	Infected	S	S	I	S	S	S	S	S	S
*S. hominis*	Non-infected	S	R	I	S	S	S	S	S	S
*S. epidermidis*	Non-infected	S	S	I	S	S	S	S	S	S
*S. capitis*	Non-infected	S	S	I	S	S	S	S	S	S
*S. aureus*	Non-infected	R	R	R	S	S	S	S	S	S
*S. caprae*	Non-infected	S	S	I	S	S	S	S	S	S
*S. aureus*	Infected	S	S	I	S	S	S	S	S	S
*S. epidermidis*	Non-infected	S	R	I	R	S	S	S	S	S

**Table 2 ijms-26-08929-t002:** *S. aureus* isolates biofilm-forming ability.

Isolates	Prosthetics Etiology	Biofilm (OD 550)	Biofilm Classification
*S. aureus*	Infected	0.51	Weak
*S. aureus*	Infected	1.02	Moderate
*S. aureus*	Non-infected	0.39	Poor
*S. aureus*	Infected	0.41	Weak
*S. aureus*	Infected	0.42	Weak
*S. aureus*	Infected	0.42	Weak
*S. aureus*	Infected	1.19	Strong
*S. aureus*	Infected	0.95	Strong
*S. aureus*	Non-infected	0.54	Moderate
*S. aureus*	Infected	1.17	Strong

**Table 3 ijms-26-08929-t003:** CoNS isolates biofilm forming ability.

Isolates	Prosthetics Etiology	Biofilm (OD 550)	Biofilm Classification
*S. epidermidis*	Infected	0.24	Poor
*S. capitis*	Infected	0.93	Moderate
*S. epidermidis*	Infected	0.42	Weak
*S. capitis*	Infected	1.03	Moderate
*S. lugdunensis*	Infected	0.41	Weak
*S. epidermidis*	Infected	0.44	Weak
*S. epidermidis*	Infected	1.06	Strong
*S. epidermidis*	Infected	0.58	Moderate
*S. epidermidis*	Non-infected	0.69	Moderate
*S. epidermidis*	Infected	1.04	Strong
*S. epidermidis*	Infected	0.91	Strong
*S. epidermidis*	Infected	0.71	Moderate
*S. capitis*	Infected	0.78	Moderate
*S. epidermidis*	Non-infected	0.56	Moderate
*S. epidermidis*	Non-infected	0.94	Strong
*S. epidermidis*	Infected	2.13	Strong
*S. lugdunensis*	Infected	1.32	Strong
*S. hominis*	Non-infected	0.89	Strong
*S. epidermidis*	Non-infected	0.15	Poor
*S. capitis*	Non-infected	0.90	Strong
*S. caprae*	Non-infected	0.73	Moderate
*S. epidermidis*	Non-infected	0.60	Moderate

**Table 4 ijms-26-08929-t004:** Concentration of c-di-GMP(pg/mL) of Staphylococcal isolates.

Isolates	Prosthetics Etiology	c-di-GMP (pg/mL)
*S. epidermidis*	Infected	477.82
*S. aureus*	Infected	366.80
*S. capitis*	Infected	239.73
*S. aureus*	Infected	450.75
*S. epidermidis*	Infected	540.38
*S. capitis*	Infected	522.06
*S. aureus*	Non-infected	513.96
*S. aureus*	Infected	448.52
*S. lugdunensis*	Infected	297.19
*S. aureus*	Infected	292.39
*S. epidermidis*	Infected	260.42
*S. aureus*	Infected	465.98
*S. epidermidis*	Infected	461.20
*S. epidermidis*	Infected	681.71
*S. aureus*	Infected	673.08
*S. epidermidis*	Non-infected	538.33
*S. epidermidis*	Infected	437.75
*S. epidermidis*	Infected	640.74
*S. epidermidis*	Infected	520.47
*S. capitis*	Infected	786.09
*S. epidermidis*	Non-infected	632.34
*S. epidermidis*	Non-infected	727.71
*S. epidermidis*	Infected	867.42
*S. aureus*	Infected	663.11
*S. lugdunensis*	Infected	482.13
*S. hominis*	Non-infected	321.20
*S. epidermidis*	Non-infected	397.97
*S. capitis*	Non-infected	496.86
*S. aureus*	Non-infected	217.72
*S.caprae*	Non-infected	871.51
*S. aureus*	Infected	430.90
*S. epidermidis*	Non-infected	247.13

**Table 5 ijms-26-08929-t005:** Presence of gene involved in biofilm formation in *S. aureus* isolates.

Isolates	Prosthetics Etiology	*icaA*Gene Presence	*fnbA*Gene Presence	*clfA*Gene Presence
*S. aureus*	Infected	Yes	No	Yes
*S. aureus*	Infected	Yes	No	Yes
*S. aureus*	Non-infected	Yes	Yes	Yes
*S. aureus*	Infected	Yes	Yes	Yes
*S. aureus*	Infected	Yes	Yes	Yes
*S. aureus*	Infected	Yes	No	Yes
*S. aureus*	Infected	Yes	Yes	Yes
*S. aureus*	Infected	Yes	Yes	Yes
*S. aureus*	Non-infected	Yes	Yes	Yes
*S. aureus*	Infected	Yes	No	Yes

**Table 6 ijms-26-08929-t006:** Presence of gene involved in biofilm formation in CoNS isolates.

Isolates	Prosthetics Etiology	*icaA*Gene Presence	*atlE*Gene Presence	*fbe*Gene Presence
*S. epidermidis*	Infected	No	Yes	Yes
*S. capitis*	Infected	Yes	No	No
*S. epidermidis*	Infected	No	Yes	Yes
*S. capitis*	Infected	No	No	No
*S. lugdunensis*	Infected	Yes	No	No
*S. epidermidis*	Infected	Yes	Yes	Yes
*S. epidermidis*	Infected	Yes	Yes	Yes
*S. epidermidis*	Infected	No	Yes	Yes
*S. epidermidis*	Non-infected	No	Yes	Yes
*S. epidermidis*	Infected	No	Yes	Yes
*S. epidermidis*	Infected	No	Yes	Yes
*S. epidermidis*	Infected	No	Yes	Yes
*S. capitis*	Infected	No	No	No
*S. epidermidis*	Non-infected	No	Yes	Yes
*S. epidermidis*	Non-infected	Yes	Yes	Yes
*S. epidermidis*	Infected	No	Yes	Yes
*S. lugdunensis*	Infected	Yes	No	No
*S. hominis*	Non-infected	No	No	No
*S. epidermidis*	Non-infected	Yes	Yes	No
*S. capitis*	Non-infected	No	No	No
*S. caprae*	Non-infected	No	No	No
*S. epidermidis*	Non-infected	No	Yes	Yes

**Table 7 ijms-26-08929-t007:** Primers used for detection of biofilm-associated genes.

Primer	Sequence (5′-3′)
icaA F	TGGCTGTATTAAGCGAAGTC
icaA R	CCTCTGTCTGGGCTTGACC
atlE F	CAACTGCTCAACCGAGAACA
atlE R	TTTGTAGATGTTGTGCCCCA
clfA F	GTAGGTACGTTAATCGGTT
clfA R	CTCATCAGGTTGTTCAGG
fnbA F	CACAACCAGCAAATATAG
fnbA R	CTGTGTGGTAATCAATGTC
fbe F	CTACAAGTTCAGGTCAAGGACAAGG
fbe R	GCGTCGGCGTATATCCTTCAG

## Data Availability

The original contributions presented in this study are included in the article. Further inquiries can be directed to the corresponding author.

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
