# Peer review of "Biofilm Formation, c-di-GMP Production, and Antimicrobial Resistance in Staphylococcal Strains Isolated from Prosthetic Joint Infections: A Pilot Study in Total Hip and Knee Arthroplasty Patients"

_ijms, 2025, doi:10.3390/ijms26188929_

Round 1
Reviewer 1 Report
Comments and Suggestions for Authors
The introduction effectively presents the general context of TJA/TJR and the problem of PJI. However, it lacks a strong, clear statement of the research gap that this study aims to address.
In line 132, the authors write that patient samples were categorized as infected or not infected based on clinical evidence. This seems too general. It would be helpful to clarify what diagnostic criteria were used for this classification.
How long were the samples stored at 4°C before culture?
It is worth providing a description of the categorization criteria used (poor, weak, moderate, strong), which the authors refer to in their research. Also missing from the description is any mention of negative control.
The crystal violet staining method is acceptable for assessing biofilm biomass, but for a more complete picture, the resazurin method is also suggested for assessing cell viability. Combining the two tests would provide more comprehensive data, which would translate into better correlation of the results with the clinical course of periprosthetic infections (PJI).
In the 2.4 Quantification of Cyclic di-GMP section, the method description is very concise and amounts to “follow the manufacturer's instructions.” Although this is often practiced, in order to ensure full reproducibility, the key steps of the extraction procedure would have to be written. The methodology also lacks information on how the c-di-GMP concentration was normalized for comparison between different isolates.
There is a contradiction in the numbers of samples with detected bacteria. The authors state that 25 samples from the infected group and 12 samples from the uninfected group showed bacterial growth. However, later in the text, the authors state that a total of 35 samples had bacteria identified.
In addition, instead of a general statement that co-infections occurred, it would be appropriate to state which specific strains occurred together and in which clinical cases.
Table 2 lists isolates such as S. epidermidis or S. aureus but lacks unique identifiers for each sample. It is also impossible to link a specific resistance profile to other data in the study, e.g., to a specific patient, to a biofilm test result, or to the presence of genes.
Table 2 contains data for only 34 isolates. Data are missing for 3 isolates, including S. lugdunensis and S. hominis, which are listed in Figure 1, as well as for isolates of other species (E. faecalis, C. striatum, M. luteus), which were intentionally excluded but inconsistently.
In Table 2, data for the antibiotics TEI (teicoplanin) and VAN (vancomycin) includes the value “N.A.” (Not Analyzed). The authors did not explain why these key antibiotics were not analysed, which significantly reduces the clinical value of this study.
Section 3.2 whether there was a statistically significant difference in biofilm-forming ability between isolates from the infected group (PJI) and isolates from the uninfected group.
Section 3.3 is incomplete. Instead of just reporting the regression result, the authors should provide detailed data to fully evaluate the role of c-di-GMP in the model studied. Without these data, the clinical and biological value of this part of the study is minimal.
Figure 4 authors indicate that a statistically significant correlation was observed between the presence of the icaA gene and c-di-GMP concentration." However, this is a comparison of means, not a correlation in the statistical sense
In line 260: "a statistically significant correlation was observed..." I think it should be: "a significant difference in c-di-GMP levels was observed..."
What is missing from the discussion is whether there was a statistically significant difference in biofilm-forming capacity or c-di-GMP concentration between isolates with PJI and isolates from the control (uninfected) group?. This thread is very important for the clinical value of the paper. Without this answer, the article is only a description of the isolates, not a study of the determinants of PJI virulence.
All figures need more descriptive and informative legends.
Line 306: The sentence "The relatively small number of isolates analyzed may also explain the lack of correlation" is a valid point, but it contradicts the previous, illogical conclusion. The authors should present this as the primary reason for the lack of a finding, rather than speculating on a connection that their data do not support.
Lines 313-317: The use of "a statistically significant linear regression ()" is immediately undermined by the low value (0.20). This suggests a weak relationship, and the authors should clarify this contradiction instead of simply presenting both facts. The value indicates that c-di-GMP is a poor predictive factor on its own.
Line 346: The term "correlation" is used again, incorrectly, to describe a statistically significant difference between two groups (strains with and without the gene). This should be changed.
The manuscript also lacks a dedicated Conclusions section.
Author Response
Comments and Suggestions for Authors – Reviewer 1
Following the IJMS article template, we moved the 'Materials and Methods' section to the end of the article body (position 4), and the 'Results' and 'Discussion' sections to positions 2 and 3, respectively.
Comment 1: The introduction effectively presents the general context of TJA/TJR and the problem of PJI. However, it lacks a strong, clear statement of the research gap that this study aims to address.
Response 1: As correctly pointed out by the Reviewer, the scope of the study was not clearly defined. Therefore, the following paragraph has been added at the end of the Introduction section to clearly state the aim of the present investigation: “The aim of this study was to characterize the microbial population identified in patients with PJI (≥ 90 days post-implantation) after TJA and TJR, focusing on c-di-GMP production and the activation of specific genes, like ica as previously described, in order to assess the actual ability of microbial populations—particularly S. Aureus and CoNS —to form biofilms. Biofilm formation represents a negative prognostic factor that significantly complicates both treatment and infection eradication. Therefore, biofilm accurate and timely identification may represent a crucial additional tool for the appropriate management of therapy, not only guiding the selection of the most effective antibiotics but also modulating treatment duration and therapeutic strategies aimed at improving infection control and patient outcomes”.
Comment 2: In line 132, the authors write that patient samples were categorized as infected or not infected based on clinical evidence. This seems too general. It would be helpful to clarify what diagnostic criteria were used for this classification.
Response 2: As correctly pointed out by the Reviewer, this sentence was too general. Therefore, we removed it from paragraph 4.2. Bacterial Isolates and Growth Conditions. To better clarify patient allocation, we further revised paragraph 4.1 Study Population as follows: “During recruitment, patients were classified as “infected” according to the Musculoskeletal Infection Society (MSIS) major criteria [42], defined by either (1) a sinus tract communicating with the joint, or (2) a positive microbiological culture from at least two separate periprosthetic tissue/fluid samples or, alternatively, if they met at least three of minor criteria”.
Comment 3: How long were the samples stored at 4°C before culture?
Response 3: We specified that adding the following paragraph: “A 1.5 mL aliquot was immediately transferred to the Microbiology Unit for analysis.”
Comment 4: It is worth providing a description of the categorization criteria used (poor, weak, moderate, strong), which the authors refer to in their research. Also missing from the description is any mention of negative control.
Response 4: A paragraph relating to thecategorization criteria has been added to the text at the 4.3 Crystal Violet Assay paragraph: “Biofilm production was categorized, based on the absorbance measured, as poor, weak, moderate, or strong, following the semi-quantitative criteria established by Di Domenico et al. (2016) [69]. Briefly, the cut-off OD (ODc) was defined as three standard deviations above the mean OD of the negative control and strains were classified as follows: OD 4 × ODc = strong biofilm producer.”. Also, regarding the description is any mention of negative control, the following description has been added: “Two-hundred µl of BHI, dispensed in triplicate, was used as a negative control.”
Comment 5: The crystal violet staining method is acceptable for assessing biofilm biomass, but for a more complete picture, the resazurin method is also suggested for assessing cell viability. Combining the two tests would provide more comprehensive data, which would translate into better correlation of the results with the clinical course of periprosthetic infections (PJI).
Response 5: We appreciate the valuable suggestion regarding the addition of the resazurin assay to assess cell viability, which would indeed provide a more comprehensive characterization of the biofilm. However, at this stage of the study, it is not feasible to implement this additional method. Moreover, performing this assay would have required prior approval from the ethics committee and the use of fresh (non-frozen) clinical isolates, which are no longer available. We will consider including this method in future studies.
Comment 6: In the 2.4 Quantification of Cyclic di-GMP section, the method description is very concise and amounts to “follow the manufacturer's instructions.” Although this is often practiced, in order to ensure full reproducibility, the key steps of the extraction procedure would have to be written. The methodology also lacks information on how the c-di-GMP concentration was normalized for comparison between different isolates.
Response 6: As correctly pointed out by the Reviewer, the following description has been added at the 4.4 Quantification of Cyclic di-GMP section: “The bacterial strain extraction procedure was performed following the instructions “B-PER® Bacterial Protein Extraction Reagent” as indicated in the Cyclic di-GMP ELISA kit document. Data analysis and normalization were performed using the Cayman spreadsheet software provided in the kit.”
Comment 7: There is a contradiction in the numbers of samples with detected bacteria. The authors state that 25 samples from the infected group and 12 samples from the uninfected group showed bacterial growth. However, later in the text, the authors state that a total of 35 samples had bacteria identified.
Response 7: As correctly pointed out by the Reviewer, we have modified the text.The data have been updated to reflect the fact that five patients dropped out. “In this study, a total of 198 synovial fluid samples were collected and sent to the Microbiology Unit at the IRCCS Azienda Ospedaliero-Universitaria of Bologna for microbiological analysis. Among these, 48 (24.2%) samples were obtained from patients with clinically suspected PJ infected etiology, and 150 (75.8%) were from prosthetics with clinically suspected non-infected etiology. Bacterial growth was detected in 24/48 samples (50%) from prosthetics with clinically suspected infection, while among the 150 clinically suspected non-infected samples, bacterial growth was observed in 9 samples (6%). Overall, in 33/198 samples (16.7%) we found bacterial growth, including 10 strains of S. aureus, 22 strains of CoNS, 1 strain of Enterococcus faecalis, 1 strain of Corynebacterium striatum, and 1 strain of Micrococcus luteus (Figure 1). Co-infections were observed in two cases: one involving S. epidermidis and E. faecalis, associated with a prosthesis with infectious etiology, and one involving a strain of S. epidermidis and a strain of Staphylococcus capitis, associated with a prosthesis with non-infectious etiology. Our study focused on Staphylococcus strains, other strains were excluded from further analysis.”
Comment 8: In addition, instead of a general statement that co-infections occurred, it would be appropriate to state which specific strains occurred together and in which clinical cases.
Response 8: As correctly pointed out by the Reviewer, the following description has been added: “Co-infections were observed in two cases: one involving S. epidermidis and E. faecalis, associated with a prosthesis with infectious etiology, and one involving a strain of S. epidermidis and a strain of Staphylococcus capitis, associated with a prosthesis with non-infectious etiology.”
Comment 9: Table 2 lists isolates such asS. epidermidisorS. aureusbutlacks unique identifiers for each sample. It is also impossible to link a specific resistance profile to other data in the study, e.g., to a specific patient, to a biofilm test result, or to the presence of genes.
Response 9: As correctly pointed out by the Reviewer, we have added the requested information to the table 1. Information relating to biofilm and c-di-GMP has been specified in additional tables (Table 2, Table 3 and Table 4) and have been included in paragraphsBiofilm forming ability (section 2.2) and c-di-GMP concentration (section 2.3) respectively.
Comment 10: Table 2 contains data for only 34 isolates. Data are missing for 3 isolates, including S. lugdunensis and S. hominis, which are listed in Figure 1, as well as for isolates of other species (E. faecalis, C. striatum, M. luteus), which were intentionally excluded but inconsistently.
Response 10: As correctly pointed out by the Reviewer,the following description has been added: “Overall, in 33/198 samples (16.7%) we found bacterial growth, including 10 strains of S. aureus, 22 strains of CoNS, 1 strain of Enterococcus faecalis, 1 strain of Corynebacterium striatum, and 1 strain of Micrococcus luteus (Figure 1). Co-infections were observed in two cases: one involving S. epidermidis and E. faecalis, associated with a prosthesis with infectious etiology, and one involving a strain of S. epidermidis and a strain of Staphylococcus capitis, associated with a prosthesis with non-infectious etiology. Our study focused on Staphylococcus strains, other strains were excluded from further analysis.”
Comment 11: In Table 2, data for the antibiotics TEI (teicoplanin) and VAN (vancomycin) includes the value “N.A.” (Not Analyzed). The authors did not explain why these key antibiotics were not analysed, which significantly reduces the clinical value of this study.
Response 11: Results for the antibiotics TEI and VAN were added to the table 1.
Comment 12: Section 3.2 whether there was a statistically significant difference in biofilm-forming ability between isolates from the infected group (PJI) and isolates from the uninfected group.
Response 12: As correctly pointed out by the Reviewer,the following description has been added: “No statistically significant difference was found between the ability to form biofilms between bacterial strains from prosthesis with infectious and non-infectious etiology (p > 0.05).”
Comment 13: Section 3.3 is incomplete. Instead of just reporting the regression result, the authors should provide detailed data to fully evaluate the role of c-di-GMP in the model studied. Without these data, the clinical and biological value of this part of the study is minimal.
Response 13: As correctly pointed out by the Reviewer, we have added the requested information to the table 4. We specified the c-di-GMP value, expressed in pg/mL, associated with each microorganism together with the etiology of the prosthesis.
Comment 14: Figure 4 authors indicate that a statistically significant correlation was observed between the presence of the icaA gene and c-di-GMP concentration." However, this is a comparison of means, not a correlation in the statistical sense
Response 14: As correctly pointed out by the Reviewer, the following description has been added: “Comparison of c-di-GMP production in icaA+ and icaA- strains. Negative strains showed significantly higher concentrations (p = 0.016), indicating multifactorial regulation of biofilm formation”.
Comment 15: In line 260: "a statistically significant correlation was observed..." I think it should be: "a significant difference in c-di-GMP levels was observed..."
Response 15: As correctly pointed out by the Reviewer, we have corrected the text.
Comment 16: What is missing from the discussion is whether there was a statistically significant difference in biofilm-forming capacity or c-di-GMP concentration between isolates with PJI and isolates from the control (uninfected) group?. This thread is very important for the clinical value of the paper. Without this answer, the article is only a description of the isolates, not a study of the determinants of PJI virulence.
Response 16: As correctly pointed out by the Reviewer, we added in the Results (section 2.2), “No statistically significant difference was found between the ability to form biofilms between bacterial strains from prosthesis with infectious and non-infectious etiology (p > 0.05).” Similarly, no statistically significant differences were observed in c-di-GMP concentrations between isolates from PJI and control prosthesis. This point is also discussed in the Discussion section, where we emphasize that although a significant correlation between biofilm formation and c-di-GMP was found, these factors alone did not discriminate between isolates from infected and non-infected prostheses.
Comment 17: All figures need more descriptive and informative legends.
Response 17: As correctly pointed out by the Reviewer, we have corrected all figures and added a more descriptive and informative legends. In particular:
- Figure 1. Distribution of bacterial isolates. Left: overall composition—coagulase‑negative staphylococci (CoNS), Staphylococcus aureus, Enterococcus faecalis, striatum and M. luteus (percentages as shown). Right (inset): breakdown of CoNS isolates only, comprising S. hominis, S. lugdunensis, S. caprae, S. epidermidis, and S. capitis.
- Figure 2. Biofilm formation ability of Staphylococcus strains evaluated by crystal violet assay. Strains show variability from poor to strong, with a prevalence of epidermidis among moderate/strong producers.
- Figure Linear regression between biofilm-forming capacity and c-di-GMP production. Despite a significant difference (p = 0.016) the low R² (R² = 0.18) indicates that other factors contribute to the biofilm production.
- Figure 4. Comparison of c-di-GMP production in icaA+ and icaA- strains. Negative strains showed significantly higher concentrations (p = 0.016), indicating multifactorial regulation of biofilm formation.
- Figure 5. Analysis of oxacillin susceptibility in fbe+ and fbe- strains. Positive strains showed a significantly higher frequency of oxacillin resistance (p = 0.031).
Comment 18: Line 306: The sentence "The relatively small number of isolates analyzed may also explain the lack of correlation" is a valid point, but it contradicts the previous, illogical conclusion. The authors should present this as the primary reason for the lack of a finding, rather than speculating on a connection that their data do not support.
Response 18: As correctly pointed out by the Reviewer, the following description has been added: “No correlation was found between the identification of methicillin-resistant isolates and their ability to produce biofilm or c-di-GMP. This lack of correlation is most likely due to the relatively small number of isolates analyzed, which limits the statistical power of our observations. Nevertheless, literature indicates that the presence of biofilm itself may offer effective protection from antibiotic action, as bacteria embedded in biofilms can survive antibiotic concentrations 10 to 10,000 times greater than their planktonic counterparts [46,52].”
Comment 19: Lines 313-317: The use of "a statistically significant linear regression (p<0.05)" is immediately undermined by the low R² value (0.20). This suggests a weak relationship, and the authors should clarify this contradiction instead of simply presenting both facts. The R² value indicates that c-di-GMP is a poor predictive factor on its own.
Response 19: As correctly pointed out by the Reviewer, the following description has been added: “Although a statistically significant linear regression (p < 0.05) was observed between biofilm production and c-di-GMP levels, consistent with findings in other bacteria such as Pseudomonas aeruginosa, Pseudomonas resinovorans, and Vibrio vulnificus [55–57], the low R² value (0.20) indicates that c-di-GMP explains only a small proportion of the variability in biofilm formation. This suggests that, while the association is real, c-di-GMP alone is a poor predictive biomarker for biofilm production and likely requires consideration in combination with other regulatory factors.”
Comment 20: Line 346: The term "correlation" is used again, incorrectly, to describe a statistically significant difference between two groups (strains with and without the fbe gene). This should be changed.
Response 20: As correctly pointed out by the Reviewer, we have corrected the textand modified it in: “could explain the statistically significant difference between the groups.”
Comment 21: The manuscript also lacks a dedicated Conclusions section.
Response 21: Thank you for your observation. We have restructured the manuscript according to the guidelines of IJMS, which require the following structure: Introduction, Results, Discussion, Materials and Methods, Conclusions. Accordingly, we have added a dedicated Conclusions section and reorganized the manuscript to comply with the journal's formatting requirements
Reviewer 2 Report
Comments and Suggestions for Authors
The manuscript “Biofilm formation, c-di-GMP production, and antimicrobial resistance in Staphylococcal Strains isolated from Prosthetic Joint Infections: a pilot study in total hip and knee arthroplasty patients” is relevant, current, and highly relevant. However, a few minor points should be clarified:
- Add the study's inclusion and exclusion criteria more clearly.
- Was a sample collected from each patient? Were "duplicate" patients excluded?
- Lines 132-133: Each sample was classified as "infected" or "not-infected" based on clinical evidence. Describe the criteria for determining whether the site was infected or not.
- Specify how long it took between collection and analysis. Remember that cooling can decrease the viability of some bacterial genera.
- Only aerobic tests were performed. This should be clear. The authors did not test for anaerobes. Please mention this in the text.
- What positive control strain for biofilm formation was used to validate the techniques? This is important. It is difficult to establish a comparative standard without a positive control. This is necessary for all experimental techniques.
- Lines 280-281: Therefore, the aim of the present study was to better characterize the virulence determinants. Okay, true...but one of the objectives was to verify the resistance determinants (which is something different from virulence). Describe this further.
Author Response
Comments and Suggestions for Authors – Reviewer 2
Following the IJMS article template, we moved the 'Materials and Methods' section to the end of the article body (position 4), and the 'Results' and 'Discussion' sections to positions 2 and 3, respectively.
The manuscript “Biofilm formation, c-di-GMP production, and antimicrobial resistance in Staphylococcal Strains isolated from Prosthetic Joint Infections: a pilot study in total hip and knee arthroplasty patients” is relevant, current, and highly relevant. However, a few minor points should be clarified:
Comment 1: Add the study's inclusion and exclusion criteria more clearly.
Response 1: In agreement with the Reviewer’s suggestion, we have provided a clearer description of the inclusion and exclusion criteria of the clinical study, as follows: “A prospective ongoing clinical study was conducted at the IRCCS Istituto Ortopedico Rizzoli in Bologna (Italy), in collaboration with the Microbiology Unit of the IRCCS Azienda Ospedaliero-Universitaria of Bologna. The study aimed to enroll patients scheduled for hip or knee prosthesis revision due to late infection (beyond 90 days) and to prosthetic failure not related to infection. The protocol was approved by the Ethics Committee of Area Vasta Emilia Centro (CE-AVEC; protocol number 37/2021/Sper/IOR) and the study was conducted in accordance with the Declaration of Helsinki (ClinicalTrial.gov: ID: NCT04858217). Written informed consent was obtained from all participants. Between February 2022 and January 2025, 203 patients were enrolled. Inclusion criteria were: 1) male and female patients aged ≥ 18 years; 2) patients scheduled for hip or knee prosthesis revision surgery after at least 90 days from the date of the primary arthroplasty due to a) late PJI, or b) non-infective causes (loosening, wear, instability, malalignment, adverse local tissue reactions, or other aseptic conditions), in patients who had not undergone previous re-operations on the same joint, and whose revision procedure was planned as a single-stage intervention; 3) availability of previous clinical data as well as laboratory and radiological examinations. Exclusion criteria were: 1) patients with early PJI, with a clinical latency of symptoms of less than 90 days; 2) patients with PJI involving joints other than the hip or knee; 3) patients with severe cognitive impairment or psychiatric disorders; 4) Pregnant women”.
Comment 2: Was a sample collected from each patient? Were "duplicate" patients excluded?
Response 2: Only one sample per patient was included in the study. Duplicate patients were excluded to ensure that each case was represented only once in the analysis.
Comment 3: Lines 132-133: Each sample was classified as "infected" or "not-infected" based on clinical evidence. Describe the criteria for determining whether the site was infected or not.
Response 3: Also in agreement with Reviewer 1’s concern on this point, we removed the sentence at lines 132–133 and provided a clearer explanation of patient allocation by adding new text to paragraph 2.1 Study Population as follows: “During recruitment, patients were classified as “infected” according to the Musculoskeletal Infection Society (MSIS) major criteria [42], defined by either (1) a sinus tract communicating with the joint, or (2) a positive microbiological culture from at least two separate periprosthetic tissue/fluid samples or, alternatively, if they met at least three of minor criteria”.
Comment 4: Specify how long it took between collection and analysis. Remember that cooling can decrease the viability of some bacterial genera.
Response 4: The following paragraph has been added : “A 1.5 mL aliquot was immediately transferred to the Microbiology Unit for analysis.”
Comment 5: Only aerobic tests were performed. This should be clear. The authors did not test for anaerobes. Please mention this in the text.
Response 5: As correctly pointed out by the Reviewer, the following description has been added: “Agar plates were incubated at 37 °C and examined for aerobic bacteria growth at 24 and 48 hours.”
Comment 6: What positive control strain for biofilm formation was used to validate the techniques? This is important. It is difficult to establish a comparative standard without a positive control. This is necessary for all esperimental techniques.
Response 6: We acknowledge the importance of using a positive control strain for validating biofilm formation techniques. However, we did not have access to a strain with a standardized biofilm-forming capacity. The evaluation of biofilm formation in our study was conducted following established protocols described in the literature (Di Domenico et al; REF 69). We recognize this as a limitation of the current study and will consider the inclusion of a validated positive control strain in future research.
Comment7: Lines 280-281: Therefore, the aim of the present study was to better characterize the virulence determinants. Okay, true...but one of the objectives was to verify the resistance determinants (which is something different from virulence). Describe this further.
Response 7: As correctly pointed out by the Reviewer, the following description has been added: “Therefore, the aim of the present study was to better characterize biofilm-associated determinants of virulence in different bacterial strains isolated from synovial fluid derived from PJIs, with particular focus on factors influencing biofilm formation rather than antimicrobial resistance mechanisms.”
Round 2
Reviewer 1 Report
Comments and Suggestions for Authors
Thank you for your response. Having reviewed the changes, I have no further comments. I believe the manuscript is now ready for publication.